# Immediate effects of visual feedback on the accuracy of foot landing adjustments in older people with diabetes mellitus: A cross-sectional study

Suzanne Martin[1]*, Simon B. Taylor[1], Shabnam Pejhan[2], Blynne L. Shideler[3], Rajna Ogrin[4], Rezaul Begg[1]

**1** Institute of Health and Sport, Victoria University, Melbourne, Victoria, Australia, **2** Department of Automotive and Mechatronics Engineering, Ontario Tech University, Oshawa, Ontario, Canada, **3** School of Medicine, Stanford University, Stanford, California, United States of America, **4** Bolton Clarke Research Institute, Melbourne, Australia

* suzanne.martin@vu.edu.au

## Abstract

Diabetes Mellitus in older adults reduces the accuracy of foot landing adjustments and increases the risk of falling. This study investigated whether targeted visual feedback enhances the accuracy of the foot landing in older participants with diabetes. Forty-eight volunteers in three groups of young, healthy older and older adults with diabetes participated. During treadmill walking, Nexus Vicon software streamed real-time orientation data of two markers on the first toes into MATLAB scripts through a Visual3D server. The system visualised real-meaningful step length or minimum toe clearance (MTC) in each step on a monitor in front of a treadmill at eye level. Participants responded to four subject-specific step-length (±10% mean baseline step length) or MTC targets (3.5 and 5.5 cm higher than the mean baseline MTC). One target was displayed every 10 steps and removed after 10 steps when participants walked without seeing any target, then a new or the same target appeared and stayed on for another 10 steps. Tasks were increasing or decreasing baseline step length or increasing the baseline MTC in response to virtual targets. The accuracy (absolute error) of each step adjustment in each 10-step block with a target was calculated. The mean accuracy of Step 1 and the mean accuracy of Steps 2–10 were measured and compared within and between groups using three-way ANOVA tests. Errors significantly differed between conditions (Step 1 and Steps 2–10) in all groups. All groups showed reduced errors in Step 1 during Steps 2–10. Among groups, the group with diabetes presented the greatest errors in Step 1 and Steps 2–10. These findings suggest that meaningful visual feedback about spatial gait parameters (step length, MTC) can improve foot-landing accuracy in older participants with diabetes, highlighting its potential as a training tool to prevent falls in this high-risk population.

**Data availability statement:** All data underlying the findings described in our paper are freely available to other researchers in a public repository based in Australia https://vuir.vu.edu.au/id/eprint/42294. Martin, Suzanne (2021) Gait Adaptability and Biofeedback in Older Adults with Diabetes. PhD thesis, Victoria University. Keywords: gait; walking; gait adaptability; older; elderly; ageing; diabetes; biofeedback; obstacle.

**Funding:** The author(s) received no specific funding for this work.

**Competing interests:** The authors have declared that no competing interests exist.

## Introduction

Diabetes mellitus increases the risk of falls in older adults due to reduced sensorimotor function [1] that reduces the accuracy of foot adjustments in response to external hazards such as obstacles [2]. It is reported that older adults with type 2 diabetes (diabetes mellitus) demonstrated worse balance, gait, and functionality, contributing to an increased fall risk compared to those without diabetes [3]. Individuals with diabetic peripheral neuropathy exhibit reduced sensory input, impaired motor control, and slower functional mobility, further elevating their risk of falling [3]. Additionally, even older adults with type 2 diabetes who do not have peripheral neuropathy may still face heightened fall risks due to subtle declines in sensory functions (somatosensory, visual, vestibular), metabolic muscle function, and cognitive performance [4]. Over 50% of older adults with diabetes report one injurious fall or two non-injurious falls annually [5]. Each year, an estimated 37.3 million falls worldwide result in injuries requiring medical attention [6]. Falls are the leading cause of injury-related hospitalisations in older Australians [7]. The most frequent injuries that result from falls are fractures followed by open wounds [7]. Fall-related injuries in people with diabetes are more serious than in those without diabetes [8]; therefore, fall prevention programs to reduce the incidence of falls in older people with diabetes are of particular importance.

Visual feedback for training more accurate foot landing adjustments to external hazards is a potential intervention to prevent falls. Targeted biofeedback was used to reduce falls in older adults and patients with chronic stroke and Parkinson's disease by training step length and minimum toe clearance (MTC) adjustments in response to virtual targets during treadmill walking [9]. Therefore, gait adaptability involves adjusting the movement of the landing foot in response to goal-oriented tasks such as stepping on projected targets [10,11]. The training program provided feedback by directly comparing foot displacement with the position of targets on the ground [10], or by comparing visualised foot displacement and targets on a monitor installed in front of the treadmill (non-immersive visual reality) [11].

Reduced accuracy of foot landing adjustments in older people has been investigated, and the varying results may reveal several methodological issues [12,13]. Despite the influence of time-constrained conditions, it is unlikely that all participants were free from age-related conditions that might affect their responses to tasks. Hence, it is important to investigate foot-landing adjustments in cohorts whose characteristics are homogenous. Aging without or with diabetes mellitus may impair vision, cognition, and peripheral nerves [14]. Reduced accuracy of foot landing adjustments may place adults with age-related conditions at increased risk of falling when negotiating unexpected hazards. When tasked to step on a virtual target and increase their step length, older adults increased the number of steps (by one), shortened their step lengths and spent more time in double support, compared with young adults [15]. This adaptation by older adults results in poorer stepping accuracy and failure to adjust steps to meet stepping targets and avoid obstacles. Similar trends have been reported in research studies with the involvement of older people with stroke and diabetes mellitus [2,16,17]. Although the development of peripheral neuropathy can be responsible for a reduced accuracy in foot landing adjustments

when responding to goal-oriented tasks in people with diabetes mellitus [11,18,19], no study has been found to investigate the effectiveness of meaningful visual feedback of spatial gait parameters in this population.

The use of visual feedback to train more accurate foot landing adjustments in individuals with diabetes mellitus without neuropathy also remained relatively unexplored; however, some investigators reported that diabetes mellitus in older people impairs the accuracy of foot landing adjustments in the sagittal plane [2]. This study aimed to present the application of a new computer system for training foot landing adjustments with visual feedback. If successful, this application could be used to prevent falls in older people living with diabetes.

The hypotheses tested in this study were (i) diabetes mellitus would impair spatiotemporal gait parameters, (ii) individuals with diabetes mellitus would be able to use visual feedback (meaningful information) and immediately increase the accuracy of their foot adjustments, and (iii) diabetes mellitus would reduce the accuracy of foot landing adjustments in an older group with diabetes compared with matched healthy older and young groups.

## Materials and methods

### Study design

A cross-sectional study design was used to investigate the effects of diabetes mellitus on foot displacement adjustments in the sagittal plane in response to visual feedback using a more developed visual feedback system [20].

### Setting

All procedures were undertaken at the Biomechanics Laboratory at Victoria University, located in metropolitan Melbourne, Australia. The Human Research Ethics Committee reviewed the proposal and approved it (HRE17-194).

### Sample size calculation

A priori power calculation was conducted using the results of a study [12] and software [21]. It was estimated that 48 participants in three groups would be required to detect an effect size of 0.66 with 95% power and a significant level of 0.05.

### Recruitment of participants

Recruitment of participants was undertaken in the western suburbs in western metropolitan Melbourne, Australia, near the university campus for ease of access to participants. Eligibility criteria included aged between 18 and 40 years for young adults and aged between 65 and 85 years for older adults; free from any musculoskeletal injury at the time of testing; free from uncorrected vision issues; able to walking without any walking aids including assistive devices; no history of falling within a year before participating in the study; free from any diabetes and ageing-related neuropathy, and able to understand English. Exclusion criteria included vision impairment (less than 20/40 based on eye chart examination); stroke; heart problems; physical injuries; a history of fall within a year before participating in the study; medications that may impact balance (antidepressants, anti-anxiety drugs, blood pressure medications, sleep aids); neuropathy (Michigan Neuropathy Screening Instrument (MNSI) [22] scores of 3 and over), and dementia (Mini-Mental State Examination (MMSE) [23] scores of less than 27).

### Experimental setup and equipment

Retroreflective markers were attached to the toe of each shoe. Toe marker trajectory data was collected by a three-dimensional motion analysis (VICON, Oxford, UK). For visualisation of real-time step length and MTC and targets on a monitor, a customised MATLAB program (MathWorks, Natick, USA) used the data by VICON streamed by a Visual3D-Server software (C-Motion, Germantown, USA). This study used a visual feedback system [20] with the addition of visualisation of the MTC in each step in the current study to enhance the accuracy of foot displacement adjustments in the sagittal plane.

## Procedure

Participants first completed a 5–10-minute treadmill walking when wearing a safety harness (familiarisation session). A preferred walking speed was determined for each participant [24]. In the baseline condition, participants walked for 10 minutes at their preferred speed without visual feedback (the monitor was off). Baseline walking trials were used to determine spatiotemporal gait parameters. Four targets were defined based on each participant's baseline step length and MTC. Two step-length targets were 10% shorter or longer than baseline mean step length and two MTC targets were 3.5 cm or 5.5 cm higher than baseline mean MTC (high MTC target and higher MTC target).

During the practice session, the real-time graphs and points of interest for step length and MTC and virtual targets were illustrated on the monitor (Fig 1). Every ten steps (resting time), one in four targets was presented on the monitor and stayed during the next ten steps. Each target was triggered three times during a total of 240 steps. However, targets were removed 12 times during 10 step-blocks of walking. In total, each participant walked without any target for 120 steps and responded to four targets during 120 steps. The ten-step blocks without any targets let participants to go back to their mean baseline step length and MTC.

## Date analysis

Only, 120 steps in response to four targets were analysed. Step adjustments were grouped into two error measures. The reactive error (Step 1) for each target was computed as the mean errors observed during the first step of each 10-step block when a target appeared in the last two steps of a 10-step block without a target. However, the online correction error (Steps 2–10) was calculated as the mean errors observed during steps 2–10, corresponding to the period when the target stayed visible, and participants used visual feedback to match their step length or MTC height with the presented target on the monitor (Fig 1). These two errors (accuracy) were compared across participants and among groups.

All statistical analyses were performed in SPSS (Version 25 for Windows, SPSS Science, Chicago, USA) at a significance level of $\alpha = 0.05$. Three-way ANOVA tests compared the gait and foot landing adjustments between groups (Group I, Group II, Group III) and between conditions (Step 1 and Steps 2–10). Nonparametric tests (Kruskal-Wallis H and Mann-Whitney U) were used when data were not normally distributed.

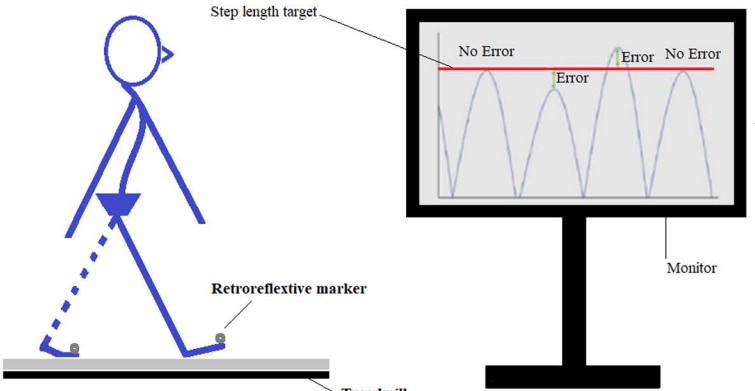

**Fig 1. Foot landing adjustments in response to short and long step length targets on a monitor.** The participant used visual feedback on the monitor to match the peaks of step-length graphs with the target (the horizontal line). The Y-axis shows the real-time length of each step and a target during four steps.

## Results

Forty-eight volunteers,16 young adults (Group I), 16 healthy older adults (Group II), and 16 older adults with diabetes (Group III) participated. Before foot landing adjustment tests, all older participants with diabetes had a mean glycated hae-moglobin A1c (GHbA1c) of 7.6%±1.8%. None of the participants had fallen in the year before their participation, and all achieved a score of 27 and over when the MMSE was completed for them. Two older adults with diabetes were excluded because they had peripheral neuropathy. The rest of the older participants achieved MNSI scores between zero and 1.5, so neuropathy was not confirmed. During treadmill familiarisation, two healthy older participants and one older participant with diabetes were excluded because they were uncomfortable walking on the treadmill.

Only collected data from 43 participants, 16 young adults (9 males, 7 females), 14 healthy older adults (8 males, 6 females), and 13 older adults with diabetes (7 males, 6 females) was used for statistical analyses. Table 1 presents participants' characteristics and baseline gait analysis.

Participants' characteristics and spatiotemporal gait parameters while walking at their preferred speeds (Table 1) were not different among groups.

In step shortening when a short step length target was presented (Fig 2), all three groups used visual feedback during the online correction (Steps 2–10) and reduced the errors in Step 1 (Z=-3.516, $p$=0.0004 in Group I, Z=-3.296, $p$=0.001 in Group II, and Z=-2.970, $p$=0.003 in Group III). Group errors were different in Step 1 and Steps 2–10 significantly (H (2) = 11.457, $p$=0.003 and H (2) = 15.467, $p$=0.0004). Errors of foot landing adjustments during step shortening were different between Group I and Group III (U=37, $p$=0.003 and U=21.5, $p$=0.0003) and Group II and Group III (U=31, $p$=0.003 and U=31, $p$=0.004).

In step lengthening when a long step length target was presented, all groups reduced their error in Step 1 during Steps 2–10 significantly (Group I, Z=-3.517, $p$=0.0004; Group II, Z=-2.480, $p$=0.013; and Group III, Z=-2.411, $p$=0.016) (Fig 3). The mean errors of the group were different in Step 1 and Steps 2–10 (H (2) = 13.201, $p$=0.001 and H (2) = 18.749, $p$=0.00009). Mean step lengthening errors were different between Group I and Group III (U=30, $p$=0.001 and U=13, $p$=0.0006) and Group II and Group III (U=36, $p$=0.008 and U=20, $p$=0.001).

In increasing MTC in response to high MTC targets (Fig 4), groups' mean errors in Step 1 reduced in Steps 2–10 significantly (Z=-2.870, $p$=0.004 in Group I, Z=-3.297, $p$=0.001 in Group II and Z=-2.760, $p$=0.006 in Group III). Groups' mean errors were different significantly in Step 1 and Steps 2–10 (H (2) = 15.599, $p$=0.0004 and H (2) = 11.081, $p$=0.004). Mean errors were different between Group I and Group III (U=22, $p$=0.0003 in Step 1 and

**Table 1. Participants' characteristics and baseline gait parameters. Young adults (Group I); healthy older adults (Group II); older adults with diabetes (Group III).**

| | Group I (n = 16) | Group II (n = 14) | Group III (n = 13) |
|---|---|---|---|
| **Participants** | 9 males, 7 females | 8 males, 6 females | 7 males, 6 females |
| **Age (years)** | 26.06±4.97 | 68.36±5.43 | 69.62±4.81 |
| **Body mass (kg)** | 75.61±9.05 | 75.04±9.75 | 76.67±11.14 |
| **Height (cm)** | 175.00±6.40 | 167.93±10.84 | 167.23±9.97 |
| **Gait velocity (m/s)** | 1.08±0.12 | 1.08±0.16 | 0.98±0.11 |
| **Stance (% of gait cycle)** | 60.71±1.32 | 60.87±1.62 | 61.83±1.81 |
| **Swing (% of gait cycle)** | 39.05±1.20 | 38.10±2.15 | 38.43±2.79 |
| **Double support (% of gait cycle)** | 10.81±1.10 | 10.83±1.12 | 11.80±1.66 |
| **Step length (% of leg length)** | 78.03±2.99 | 77.37±6.24 | 73.21±6.99 |

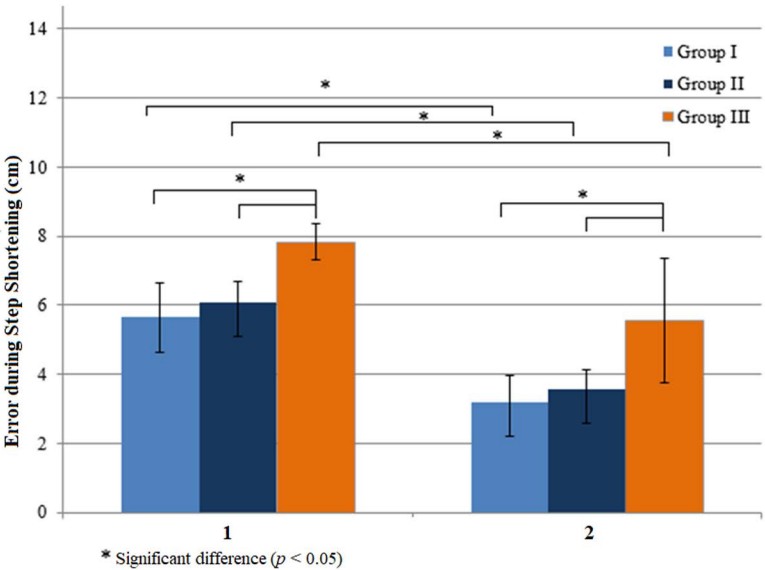

**Fig 2. Effects of visual feedback on errors during step shortening.** The errors in Step 1 (1) were significantly greater than those in Steps 2-3 (2). Young adults (Group I); healthy older adults (Group II); older adults with diabetes (Group III).

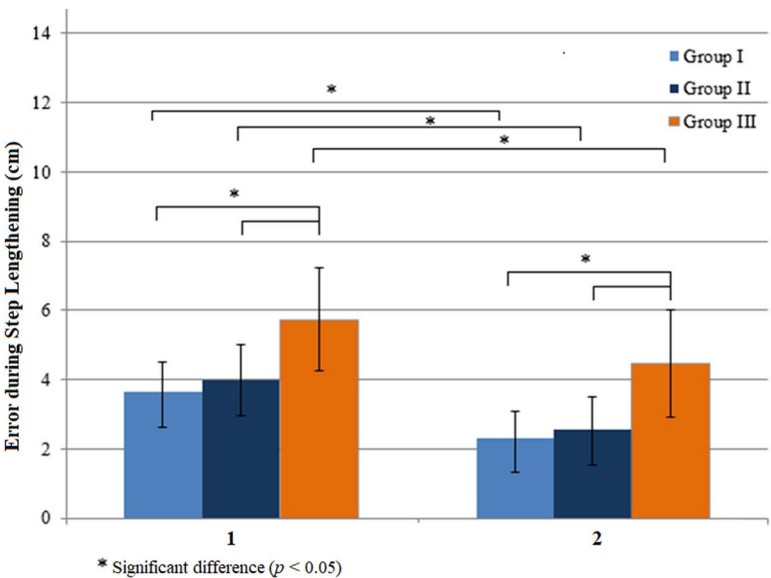

**Fig 3. Effects of visual feedback on errors during step lengthening.** Errors in Step 1 (1) were significantly reduced in Steps 2-3 (2). Young adults (Group I); healthy older adults (Group II); older adults with diabetes (Group III).

U = 37.5, $p = 0.004$ in Steps 2–10) and Group II and Group III (U = 31, $p = 0.004$ in Step 1 and U = 42, $p = 0.017$ in Steps 2–10).

During increasing MTCs in response to MTC targets that were 5.5 cm higher than baseline MTCs (Fig 5), participants in all groups used biofeedback to reduce the errors in Step 1 during Steps 2–10 (Z = -3.517, $p = 0.0004$ in Group I, Z = -3.234, $p = 0.001$ in Group II, Z = -2.691, $p = 0.007$ in Group III). Mean errors were also different between groups in Step

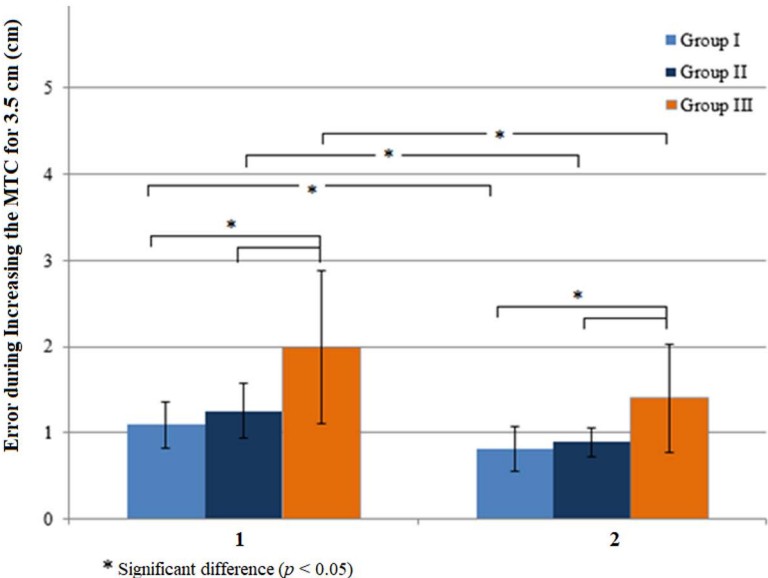

**Fig 4. Effects of visual feedback on errors during increasing the minimum toe clearance (MTC) when high MTC targets that were 3.5 cm higher than baseline MTC were presented.** The errors in Step 1 (1) were significantly reduced in Steps 2-3 (2). Group I (young adults), Group II (healthy older adults), and Group III (older adults with diabetes).

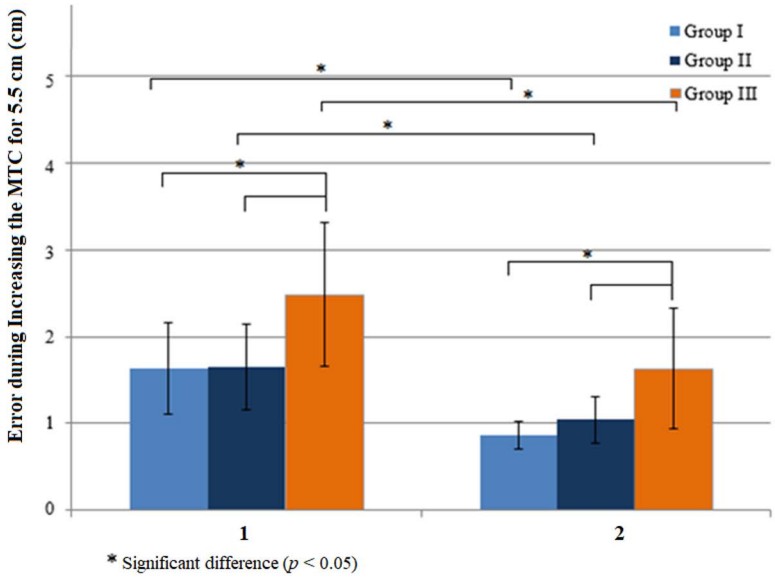

**Fig 5. Effects of visual feedback on errors during increasing the minimum toe clearance (MTC) when higher MTC targets that were 5.5 cm higher than baseline MTC were presented.** The errors in Step 1 (1) were significantly reduced in Steps 2-3 (2). Young adults (Group I); healthy older adults (Group II); older adults with diabetes (Group III).

1 and Steps 2–10 (H (2) = 10.846, $p = 0.004$ and H (2) = 12.563, $p = 0.002$), with significant differences between Group I and Group III (U = 35, $p = 0.002$ with biofeedback and U = 33, $p = 0.002$ without biofeedback) and Group II and Group III (U = 35.5, $p = 0.007$ with biofeedback and U = 41.5, $p = 0.016$ without biofeedback).

## Discussion

This study found that older participants with diabetes mellitus had the most inaccurate foot landing in response to displayed virtual targets when compared to older and younger participant groups. However, older participants with diabetes were able to use meaningful information about the difference between their actual step length/minimum toe clearance and virtual targets to reduce the differences between groups on their performance (adjusted step length and MTC) and virtual targets during the online correction.

The findings supported the first two hypotheses, highlighting the role of visual feedback in gait adaptation among participants. In line with previous research [24,25], gait characteristics during walking at preferred speeds, when no challenge was involved (baseline gait analysis), did not reveal significant differences between groups. Bard et al. [25] reported that when a target suddenly appeared on the monitor, the response was reactive and did not require further processing of afferent (sensory) feedback. However, when the target stayed visible for a few steps, it provided some feedback about the foot landing in the previous step. Therefore, a feedforward correction model was used to improve foot landing adjustment in the following steps. Consistent with previous research on foot adjustments in older adults and stroke survivors [15,17,26], this study found that receiving detailed and meaningful information about a specific task - e.g., walking with long steps - improved the accuracy of foot landing in older adults with diabetes. Like young and healthy older adults, the older participants with diabetes compared their step lengths and MTC heights with presented targets to reduce errors. Increased errors of foot displacement adjustments in older participants with diabetes might result from the delayed comparison between the expected and the real performance during the online correction (Steps 2–10) [27] or the instantaneous comparison between the goal and expected sensory feedback during inter-sensory and visuomotor integration [28]. The corticospinal system plays a key role in gait adaptation when visual feedback is available, as the visual input excites corticospinal neurons in the motor cortex and along the corticospinal pathway [29,30]. This influences neural activity measured by electroencephalography while walking with short steps [31]. Given this involvement, the reduced errors observed during steps 2–10 in older participants with diabetes in comparison with their errors in Step 1 may reflect corticospinal contributions to gait adjustments. During steps 2–10, the participants made voluntary adjustments to their performance based on feedback. Older participants with diabetes might use a similar forward model [27] in other groups of participants to compare their foot landing response with the presented target, which helped them refine the accuracy of the internal forward model in their next step/response. This period involved decision-making and reprogramming of the performed task in the following step [32]. Hence, the forward model enabled the central nervous system to predict the errors of motor commands by modulating feedback loops. In line with previous research [33], the participants reduced their foot landing errors in response to targets during the online correction (Steps 2–10). However, increased standard deviation of foot landing errors during walking with feedback may indicate an increased risk of falling [34]. The combined effects of two independent variables (diabetes and ageing) may lead to impairment of the central nervous system, which is responsible for processing and integrating information [35,36]. This impairment in the integrated brain and spinal neural network activity during walking increases the latencies of evoked potentials and reduces conduction velocity in peripheral nerves [37]. Consequently, older participants with diabetes may need more time to respond during the online correction phase (Steps 2–10) compared with participants without diabetes. Since the presented tasks had similar available times for all participants, errors of foot adjustments for online correction (Steps 2–10) increased in older participants with diabetes.

The finding also supported the third hypothesis - diabetes mellitus would decrease the accuracy of foot landing adjustments in older participants with diabetes. The older participants with diabetes showed reduced accuracy of foot adjustments when targets in Step 1, compared with the healthy older and young participants. The older participants with diabetes showed higher mean errors compared to healthy older and younger groups. Previous studies [38,39] identified that older adults may encounter more challenges in modulating and adapting foot trajectory with a target that shifts forward, backward, or upward. Additionally, the older participants with diabetes in our study displayed increased errors during online correction (Steps 2–10) compared with the other groups. Notably, this impairment was not observed in

older participants without diabetes, suggesting that these deficits may be attributed to diabetes-related factors rather than age-related changes. Dingwell et al. [40] also reported that diabetes-related impairments can increase the physiological noise of the central nervous system. Also, the frontal cortex executes response inhabitation which is important to avoid falling by stopping ongoing commands and modulating them based on sensory information [41,42]. Hence, the increased errors in foot-landing adjustments in older participants with diabetes in the current study may be explained by several factors, including frontal and parietal cortex lesions [43], lateralised right hemisphere [44], and reduced abilities to control the development of force in response to tasks [39]. It is important to highlight that these underlying mechanisms have been studied in individuals with Parkinson's disease and those who have experienced a stroke. Therefore, further investigation is needed to assess the effect of these mechanisms on individuals with diabetes mellitus.

The higher error during the online correction (Steps 2–10) in the older group with diabetes compared with the control groups (younger and healthy older adult participants) may also suggest the presence of learning deficits. In a study on diabetic rats, learning deficits were associated with changes in the hippocampus that were dependent on the severity of diabetes [45]. However, it is important to note that the older participants with diabetes were able to decrease their errors during the online correction (Steps 2–10) compared to their foot adjustments without feedback (Step 1), indicating that any learning impairment among participants was minimal and negligible. This suggests that more practice and exposure to feedback can improve performance [46].

This study examined the effectiveness of a developed visual feedback system for foot landing adjustments in older adult participants with diabetes in a safe, controlled environment, marking it as the first investigation of its kind. However, the study also encountered some limitations. The system was limited to quantifying the accuracy of foot landing adjustments in the sagittal plane; therefore, foot adjustments were not assessed in the mediolateral direction. In future research, it would be valuable to add MATLAB scripts that enable assessment of the foot landing adjustments in the mediolateral direction. Another limitation of this study was the exclusion of participants with a history of falls and neuropathy. These individuals may be able to increase the accuracy of foot landing adjustments using visual feedback. Future research will also investigate the long-term effects of visual feedback for more accurate foot-landing adjustments.

## Conclusions

This study investigated the application of a novel visual feedback system for training foot-landing adjustments in older participants with diabetes mellitus. The findings demonstrated that the visual feedback system used in this study significantly enhanced the accuracy of sagittal foot landing adjustments of the leading foot in older participants with diabetes. Specifically, older participants with diabetes mellitus adjusted their step length and minimum toe clearance more accurately when provided with real-time visual feedback using the role of feedforward biofeedback during step adaptation in the sagittal plane. The findings also highlighted the potential of visual feedback as a training tool to enhance gait control, which may contribute to fall prevention in this high-risk population.

## Author contributions

**Conceptualization:** Suzanne Martin, Simon B. Taylor, Rajna Ogrin, Rezaul Begg.

**Formal analysis:** Suzanne Martin, Simon B. Taylor.

**Investigation:** Suzanne Martin, Simon B. Taylor, Rezaul Begg.

**Methodology:** Suzanne Martin, Blynne L. Shideler, Rezaul Begg.

**Project administration:** Suzanne Martin.

**Resources:** Suzanne Martin, Simon B. Taylor, Shabnam Pejhan, Rezaul Begg.

**Software:** Simon B. Taylor, Blynne L. Shideler.

**Supervision:** Simon B. Taylor, Rajna Ogrin, Rezaul Begg.

**Validation:** Suzanne Martin, Simon B. Taylor, Blynne L. Shideler.

**Visualization:** Suzanne Martin, Shabnam Pejhan, Blynne L. Shideler.

**Writing – original draft:** Suzanne Martin, Shabnam Pejhan.

**Writing – review & editing:** Suzanne Martin, Simon B. Taylor, Shabnam Pejhan, Blynne L. Shideler, Rajna Ogrin, Rezaul Begg.

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
