## [Decision Letter · Decision Letter 0]

6 Feb 2025

PONE-D-24-44198Immediate Effects of Visual Feedback on the Accuracy of Foot Landing Adjustments in Older People with diabetes mellitusPLOS ONE

Dear Dr. Martin,

Thank you for submitting your manuscript to PLOS ONE. After careful consideration, we feel that it has merit but does not fully meet PLOS ONE’s publication criteria as it currently stands. Therefore, we invite you to submit a revised version of the manuscript that addresses the points raised during the review process.

We look forward to receiving your revised manuscript.

Kind regards,

Renato S. Melo, PhD

Academic Editor

PLOS ONE

Journal Requirements:

2. In the online submission form, you indicated that the whole data is available through a request via Victoria University Repository. 

Reviewers' comments:

Reviewer's Responses to Questions

**Comments to the Author**

1. Is the manuscript technically sound, and do the data support the conclusions?

Reviewer #1: Partly

Reviewer #2: Yes

Reviewer #3: Partly

2. Has the statistical analysis been performed appropriately and rigorously? 

Reviewer #1: Yes

Reviewer #2: I Don't Know

Reviewer #3: I Don't Know

3. Have the authors made all data underlying the findings in their manuscript fully available?

Reviewer #1: Yes

Reviewer #2: Yes

Reviewer #3: Yes

4. Is the manuscript presented in an intelligible fashion and written in standard English?

Reviewer #1: No

Reviewer #2: Yes

Reviewer #3: Yes

5. Review Comments to the Author

Reviewer #1: General comments

Thank you for giving me the opportunity to review this manuscript. The scope of the study is relevant to the readers of PlosOne. The study is interesting, but the manuscript has flaws which must be addressed before it can be suitable for publication. My main concern is the lack of a specified aim and that the method section is hard to follow. Please find my specific comments below.

Title

I suggest adding the design of the study to the title

Abstract

Baseline differences between the groups can be deleted, it is obvious that the group with diabetes differed in that aspect from the other groups and from the younger group in the aspects of diabetes and age.

I acknowledge that there is a word limit in abstract, but I still recommend the authors to include conclusion in the abstract.

Introduction

Line 55: Ref 5 is used for the statement that around 30-50% of falls cause minor injuries, this is not correct, ref 5 concerns mortality rates after fracture.

Line 68-70: please add a reference.

Line 75-77: The sentence “Compared with young adults, older people increase one shorter step and spent more time in double support when responding to an obstacle and a long step length target”. This sentence is hard to understand, what do you want to say with the sentence? I had to read the reference to be able to understand.

Line 86-87: The sentence “Therefore, this study used a more developed visual feedback system to enhance the accuracy of foot displacement adjustments in the sagittal plane.” This should be moved to method.

Aim

Aim is lacking, there are only hypotheses. Please add an aim for the study.

Materials and methods

All information is given in a single paragraph, and all information is mixed. I strongly advice the authors to structure the materials and methods section. Start with stating the design of the study. Gather all information about setting in one paragraph, participants in one paragraph, procedure in one etc.

Results

Line 148-149: information that data collection was performed at the Biomechanis laboratory at Victoria University should be moved to methods (settings).

Line 147-148 and line 157-158: the same information.

Line: 165: Delete “Mean”.

Discussion

What seems to be an aim is stated in the first sentence. The first paragraph in the discussion preferably includes main findings.

Line 239-242: It is very difficult to understand what you mean with the sentence starting with “When visual feedback is received”. Please rephrase. Also, this information (presented in a way that is understandable) should be given before the sentence “”Furthermore, reduced errors during steps 2-10 in the group of older adult with diabetes compared with errors in Step 1 can be related to corticospinal involvement”.

Line 291 and line 301: the word “elderly” is considered as ageism and should be avoided. Please check the entire manuscript. Don’t call me “old”: Avoiding ageism when writing about aging | National Institute on Aging

Conclusion

Conclusion should relate to the aim. Please check the conclusion after you have added an aim. Also move the sentence about implications for future research to the end of the discussion.

Tables and figures

Table 1: add “gait” to velocity.

References

Ref 5 is not adequat.

Reviewer #2: Thank for inviting me to review this manuscript. The authors explore the impact of visual feedback on the accuracy of foot landing adjustments in older diabetes patients. The focus on sensory motor functions in older adults with diabetes addresses a critical topic, as impaired functions increases the risk of falling in this population.

Overall, the manuscript is well-structured, with robust methods and a detailed discussion and conclusion. The experimental hypothesis,design and method is robust, with clear group stratification, the inclusion and exclusion criteria are thorough and justified. The manuscript does discuss the probable mechanisms underlying the observed impairments in visual feedback processing, particularly the potential contributions of peripheral neuropathy, proprioceptive deficits, or cognitive-motor interference.

The manuscript could benefit from providing a legend or description of the labels Z, U, and H in the figures to enhance clarity. The figures may be more informative and self explaining.

Line 123: The U value of both group 2 and 3 is 31.Is this an error?

While the research ensured enhanced internal validity, they may limit generalizability to individuals with more severe comorbidities or varying durations of diabetes. I recommend this article may be accepted with minor revision

Reviewer #3: Thank you for the opportunity to review this article. My thoughts on it are listed below:

- I missed the authors mentioning the prevalence of falls in the study population. Therefore, I would like a paragraph to be created in the discussion or introduction mentioning this outcome. Therefore, I suggest two references below so that the authors can create this paragraph in the study.

Hewston P, Deshpande N. Falls and Balance Impairments in Older Adults with Type 2 Diabetes: Thinking Beyond Diabetic Peripheral Neuropathy. Can J Diabetes. 2016;40(1):6-9.

Tavares NMB, Silva JM, Silva MDMD, et al. Balance, Gait, Functionality and Fall Occurrence in Adults and Older Adults with Type 2 Diabetes Mellitus and Associated Peripheral Neuropathy. Clin Pract. 2024;14(5):2044-2055.

6. PLOS authors have the option to publish the peer review history of their article (what does this mean? ). If published, this will include your full peer review and any attached files.

**Do you want your identity to be public for this peer review?** For information about this choice, including consent withdrawal, please see our Privacy Policy .

Reviewer #1: No

Reviewer #2: No

Reviewer #3: No

---

## [Author Response · Author response to Decision Letter 1]

22 Mar 2025

Reviewers' comments:

Reviewer #1

General comments: Thank you for giving me the opportunity to review this manuscript. The scope of the study is relevant to the readers of PlosOne. The study is interesting, but the manuscript has flaws which must be addressed before it can be suitable for publication. My main concern is the lack of a specified aim and that the method section is hard to follow. Please find my specific comments below.

Response: Thank you for your comment. We have revised the introduction section to clarify the aim of our study. Additionally, the methods section has been updated to enhance the flow and clarity of the methodology.

Comment: Title: I suggest adding the design of the study to the title

Response: Thank you for your comment. The title “Immediate effects of visual feedback on the accuracy of foot landing adjustments in older people with diabetes mellitus” was changed to “Immediate effects of visual feedback on the accuracy of foot landing adjustments in older people with diabetes mellitus: A cross-sectional study”.

Comment: Abstract: Baseline differences between the groups can be deleted, it is obvious that the group with diabetes differed in that aspect from the other groups and from the younger group in the aspects of diabetes and age. I acknowledge that there is a word limit in abstract, but I still recommend the authors to include conclusion in the abstract.

Response: The abstract has been revised following the reviewer’s recommendations. We also added a summary of the conclusion at the end of the abstract (lines 49-52 of the revised manuscript). “These findings suggest that meaningful visual feedback about spatial gait parameters (step length, MTC) can improve foot-landing accuracy in older participants with diabetes, highlighting its potential as a training tool to prevent falls in this high-risk population.” Additionally, we revised the abstract to ensure that the word limit is adhered to as closely as possible (298 words).

Comment: Introduction: Line 55: Ref 5 is used for the statement that around 30-50% of falls cause minor injuries, this is not correct, ref 5 concerns mortality rates after fracture.

Response: Thank you for your attention. We checked Ref 5 and added, “Each year, an estimated 37.3 million falls worldwide result in injuries requiring medical attention [6]. Falls are the leading cause of injury-related hospitalisations in older Australians [7]. The most frequent injuries that result from falls are fractures followed by open wounds [7]” in lines 66-69 of the revised manuscript and added their references at the end of the manuscript:

“[6] World Health Organization. (n.d.). Falls 2021. https://www.who.int/news-room/fact-sheets/detail/falls (accessed February 9, 2025).

[7] Australian Institute of Health and Welfare. Falls in older Australians 2019–20: hospitalisations and deaths among people aged 65 and over. 2022.”

Also a reference is added to line 71:

“[8] Diabetes: Australian facts 2020. https://www.aihw.gov.au/reports/diabetes/diabetes (accessed March 9, 2025).”

Comment: Line 68-70: please add a reference.

Response: Thank you for your comment. References 12 and 13 have been added in line 83 of the revised manuscript and at the end:

“[12] Liu M-W, Hsu W-C, Lu T-W, Chen H-L, Liu H-C. Patients with type II diabetes mellitus display reduced toe-obstacle clearance with altered gait patterns during obstacle-crossing. Gait Posture 2010;31:93–9. https://doi.org/https://doi.org/10.1016/j.gaitpost.2009.09.005.

[13] Hsu W-C, Liu M-W, Lu T-W. Biomechanical risk factors for tripping during obstacle—Crossing with the trailing limb in patients with type II diabetes mellitus. Gait Posture 2016;45:103–9.”

Comment: Line 75-77: The sentence “Compared with young adults, older people increase one shorter step and spent more time in double support when responding to an obstacle and a long step length target”. This sentence is hard to understand, what do you want to say with the sentence? I had to read the reference to be able to understand.

Response: Thank you for your comments. Please find the revised section in lines 89-91 of the revised manuscript: “When tasked to step on a virtual target and increase their step length, older adults increased the number of steps (by one), shortened their step lengths and spent more time in double support, compared with young adults”

Also, a few edits are applied in lines 92-95 to improve clarity. Following is the revised text:

“This adaptation by older adults results in poorer stepping accuracy and failure to adjust steps to meet stepping targets and avoid obstacles”

Comment: Line 86-87: The sentence “Therefore, this study used a more developed visual feedback system to enhance the accuracy of foot displacement adjustments in the sagittal plane.” This should be moved to method.

Response: This section was moved to lines 115-117 of the revised manuscript in Materials and Methods.

Comment: Aim: Aim is lacking, there are only hypotheses. Please add an aim for the study.

Response: The aim is added in lines 105-107 of the revised manuscript: “This study aimed to present the application of a new computer system for training foot landing adjustments with visual feedback. If successful, this application could be used to prevent falls in older people living with diabetes.”

Comment: Materials and methods: All information is given in a single paragraph, and all information is mixed. I strongly advice the authors to structure the materials and methods section. Start with stating the design of the study. Gather all information about setting in one paragraph, participants in one paragraph, procedure in one etc.

Response: We thank you for your great comment. We used the opportunity to reorganise Material and Methods. The revised structure is as follows:

Study design (line 114-117), Setting (line 125-128), Sample size calculation (line 129-132), Recruitment of participants (133-147), Experimental setup and equipment (line 158-165), Procedure (line 166-181), and Data analysis (line 182-202)

The content was also edited to improve the clarity of the text. Thank you again for your comment.

Comment: Results: Line 148-149: information that data collection was performed at the Biomechanis laboratory at Victoria University should be moved to methods (settings).

Response: Thank you for your comment. Please find the revision in lines 204-206 of revised manuscript: “Forty-eight volunteers,16 young adults (Group I), 16 healthy older adults (Group II), and 16 older adults with diabetes (Group III) participated.”

Comment: Line 147-148 and line 157-158: the same information.

Response: This section has been revised to clarify that the number of participants included in the statistical analysis differs from the number who completed their data collection sessions. In total, five participants were excluded during the further assessment for not meeting the selection criteria or for other valid justifications that can be found in lines 209-213: “Two older adults with diabetes were excluded because they had peripheral neuropathy. The rest of the older participants achieved MNSI scores between zero and 1.5, so neuropathy was not confirmed. During treadmill familiarisation, two healthy older participants and one older participant with diabetes were excluded because they were uncomfortable walking on the treadmill.”

We also elaborated it in lines 214-218 of the revised manuscript: “Collected data collected from 43 participants, 16 young adults (9 males, 7 females), 14 healthy older adults (8 males, 6 females), and 13 older adults with diabetes (7 males, 6 females) was used for statistical analysis.”

Comment: Line: 165: Delete “Mean”.

Response: “Mean” was deleted in the revised manuscript, line 223.

Comment: Discussion: What seems to be an aim is stated in the first sentence. The first paragraph in the discussion preferably includes main findings.

Response: Thank you for your invaluable comment. The aim was removed and added to the introduction (lines 102-105 of revised manuscript). The main findings were included in the first paragraph of Discussion, lines 276-281.

“This study found that older participants with diabetes mellitus had the most inaccurate foot landing in response to displayed virtual targets when compared to older and younger participant groups. However, older participants with diabetes were able to use meaningful information about the difference between their actual step length/minimum toe clearance and virtual targets to reduce the differences between groups on their performance (adjusted step length and MTC) and virtual targets during the online correction.”

Accordingly, the second paragraph of discussion was edited in lines 286-292 to link two paragraphs “The findings supported the first two hypotheses, highlighting the role of visual feedback in gait adaptation among participants. In line with previous research [24,25], gait characteristics during walking at preferred speeds, when no challenge was involved (baseline gait analysis), might not reveal significant differences between groups.”

Comment: Line 239-242: It is very difficult to understand what you mean with the sentence starting with “When visual feedback is received”. Please rephrase. Also, this information (presented in a way that is understandable) should be given before the sentence “”Furthermore, reduced errors during steps 2-10 in the group of older adult with diabetes compared with errors in Step 1 can be related to corticospinal involvement”.

This is revised to clarify the discussion (Lines 306-312 of revised manuscript). The following is the revised version:

“The corticospinal system plays a key role in gait adaptation when visual feedback is available, as the visual input excites corticospinal neurons in the motor cortex and along the corticospinal pathway [28,29]. This influences neural activity measured by electroencephalography while walking with short steps [30]. Given this involvement, the reduced errors observed during steps 2–10 in older participants with diabetes, compared to errors in Step 1, may reflect corticospinal contributions to gait adjustments”

Comment: Line 291 and line 301: the word “elderly” is considered as ageism and should be avoided. Please check the entire manuscript. Don’t call me “old”: Avoiding ageism when writing about aging | National Institute on Aging

Response: Thank you for your reminder. We changed the word “elderly” twice in our manuscript where it was used mistakenly. However, we decided to leave reference number 14 with the term of “elderly” in its title.

Comment: Conclusion: Conclusion should relate to the aim. Please check the conclusion after you have added an aim. Also move the sentence about implications for future research to the end of the discussion.

Response: Thank you for your comment. We used your feedback and rewrite the conclusion related to the aim.

lines 377-386: “This study investigated the application of a novel visual feedback system for training foot-landing adjustments in older participants with diabetes mellitus. The findings demonstrated that the visual feedback system used in this study significantly enhanced the accuracy of sagittal foot landing adjustments of the leading foot in older participants with diabetes. This supports the role of feedforward biofeedback in gait adaptation. Specifically, older participants with diabetes mellitus adjusted their step length and minimum toe clearance more accurately when provided with real-time visual feedback. These results highlight the potential of visual feedback as a training tool to enhance gait control, which may contribute to fall prevention in this high-risk population.”

Also, we moved the implications to the end of the discussion.

lines 373-374 of revised manuscript: “Future research will also investigate the long-term effects of visual feedback for more accurate foot-landing adjustments.”

Comment: Tables and figures: Table 1: add “gait” to velocity.

Response: “Velocity” was changed to “Gait velocity” in Table 1.

Comment: References: Ref 5 is not adequate.

Response: Thank you for your comment. We replaced Ref 5. We changed “(Goldacre, et al., 2002)” to a more adequate reference “(Roman de Mettelinge, et al., 2013)” in References (line 405)

Reviewer #2

Comment: The manuscript could benefit from providing a legend or description of the labels Z, U, and H in the figures to enhance clarity. The figures may be more informative and self explaining.

Response: Thank you for your comment. We used legend and description of the labels in each figure. For example, in Fig 2, legends Group I, Group II, and Group III were used and described in Fig1’s caption. We used a consistent method to illustrate the results in our manuscript.

Comment: Line 123: The U value of both group 2 and 3 is 31.Is this an error?

Response: Thank you for your attention. We checked the original data analyses in SPSS to ensure that Line 231 in the Revised Manuscript with Track Changes did not include any errors. Probably because the distribution of data in the two groups differed in spread and shape but the groups’ medians were very similar, U values became the same. We did not change anything in Line 231.

Comment: While the research ensured enhanced internal validity, they may limit generalizability to individuals with more severe comorbidities or varying durations of diabetes. I recommend this article may be accepted with minor revision

Response: Thank you for your insights. We admit that we can’t generalise these results and use the results in older people with diabetic peripheral neuropathy who were excluded from participating in our research project. We already mentioned this limitation in lines 370-371. “Another limitation of this study was the exclusion of participants with a history of falls and neuropathy.” Therefore, we did not change anything in these lines.

Reviewer #3

Comment: I missed the authors mentioning the prevalence of falls in the study population. Therefore, I would like a paragraph to be created in the discussion or introduction mentioning this outcome. Therefore, I suggest two references below so that the authors can create this paragraph in the study.

Hewston P, Deshpande N. Falls and Balance Impairments in Older Adults with Type 2 Diabetes: Thinking Beyond Diabetic Peripheral Neuropathy. Can J Diabetes. 2016;40(1):6-9.

Tavares NMB, Silva JM, Silva MDMD, et al. Balance, Gait, Functionality and Fall Occurrence in Adults and Older Adults with Type 2 Diabetes Mellitus and Associated Peripheral Neuropathy. Clin Pract. 2024;14(5):2044-2055.

Response: Thank you for your comment. Although the focus of the study wasn’t older adults with diabetic peripheral neuropathy, we used your recommended references and made some adjustments in lines 58-65 of the revised manuscript as follows:

“It is reported that older adults with type 2 diabetes (diabetes mellitus) demonstrated worse balance, gait, and functionality, contributing to an increased fall risk compared to those without diabetes [3]. Individuals with diabetic peripheral neuropathy exhibit reduced sensory input, impaired motor control, and slower functional mobility, further elevating their risk of falling [3]. Additionally, even older adults with type 2 diabetes who do not have peripheral neuropathy may still face heightened fall risks due to subtle declines in sensory functions (somatosensory, visual, and vestibular), metabolic muscle function, and cognitive performance [4].”

The following references were added at the end of the manuscript:

“[3] Tavares NMB, Silva JM, Silva MDM da, Silva LDT, Souza JN de, Ithamar L, et al. Balance, Gait, Functionality and Fall Occurrence in Adults and Older Adults with Type 2 Diabetes Mellitus and Associated Peripheral Neuropathy. Clin Pract 2024;14.

[4] Hewston P, Deshpande N. Falls and Balance Impairments in Older Adults with Type 2 Diabetes: Thinking Beyond Diabetic Peripheral Neuropathy. Can J Diabetes 2016;40:6–9. https://doi.org/https://doi.org/10.1016/j.jcjd.2015.08.005.”

---

## [Decision Letter · Decision Letter 1]

10 Apr 2025

Immediate Effects of Visual Feedback on the Accuracy of Foot Landing Adjustments in Older People with Diabetes Mellitus: A Cross-Sectional Study.

PONE-D-24-44198R1

Dear Dr. Martin,

We’re pleased to inform you that your manuscript has been judged scientifically suitable for publication and will be formally accepted for publication once it meets all outstanding technical requirements.

Kind regards,

Renato S. Melo, PhD

Academic Editor

PLOS ONE

Additional Editor Comments (optional):

Reviewers' comments:

Reviewer's Responses to Questions

**Comments to the Author**

1. If the authors have adequately addressed your comments raised in a previous round of review and you feel that this manuscript is now acceptable for publication, you may indicate that here to bypass the “Comments to the Author” section, enter your conflict of interest statement in the “Confidential to Editor” section, and submit your "Accept" recommendation.

Reviewer #2: All comments have been addressed

Reviewer #3: All comments have been addressed

2. Is the manuscript technically sound, and do the data support the conclusions?

Reviewer #2: Yes

Reviewer #3: Yes

3. Has the statistical analysis been performed appropriately and rigorously? 

Reviewer #2: I Don't Know

Reviewer #3: I Don't Know

4. Have the authors made all data underlying the findings in their manuscript fully available?

Reviewer #2: Yes

Reviewer #3: Yes

5. Is the manuscript presented in an intelligible fashion and written in standard English?

Reviewer #2: Yes

Reviewer #3: Yes

6. Review Comments to the Author

Reviewer #2: (No Response)

Reviewer #3: The authors did a good job. All my requests were met by the authors and modified in the manuscript. I believe that this latest version is clearer and better written for better understanding by the authors. Therefore, I believe that this latest version of the manuscript is ready to be accepted for publication.

7. PLOS authors have the option to publish the peer review history of their article (what does this mean? ). If published, this will include your full peer review and any attached files.

**Do you want your identity to be public for this peer review?** For information about this choice, including consent withdrawal, please see our Privacy Policy .

Reviewer #2: No

Reviewer #3: No

---

## [Editor Report · Acceptance letter]

PONE-D-24-44198R1

PLOS ONE

Dear Dr. Martin,

I'm pleased to inform you that your manuscript has been deemed suitable for publication in PLOS ONE. Congratulations! Your manuscript is now being handed over to our production team.

Kind regards,

on behalf of

Dr. Renato S. Melo

Academic Editor

PLOS ONE